# Self-efficacy, self-esteem, and happiness in older adults: A cross-sectional study

**Towhid Babazadeh[1], Soheila Ranjbaran[1], Sara Pourrazavi[2], Khalil Maleki Chollou◉[3]\*, Akbar Nadi[4], Erfan Saeedi Tazekand[4]**

1 Department of Public Health, Sarab Faculty of Medical Sciences, Sarab, Iran, 2 Research Center of Psychiatry and Behavioral Sciences, Tabriz University of Medical Sciences, Tabriz, Iran, 3 Department of Nursing, Sarab Faculty of Medical Sciences, Sarab, Iran, 4 Student Research Committee, Sarab Faculty of Medical Sciences, Sarab, Iran

\* khmaleki444@gmail.com

## Abstract

### Background

Happiness is crucial for well-being in older people, but it can be challenged by various health issues. While previous research has explored individual predictors of happiness, there is limited understanding of how self-efficacy—the belief in one's ability to manage challenges—and self-esteem, or positive self-regard, together influence happiness in elderly populations. Hence, this research aimed to fill this gap by investigating the association of self-efficacy and self-esteem with happiness in elders.

### Methods

A cross-sectional study was conducted with 400 individuals aged 60 years or older who visited health centers in Sarab, Iran, from April to June 2023. Data were collected using valid and reliable instruments, including the Oxford Happiness Questionnaire, Sherer's Self-Efficacy Scale, and the Rosenberg Self-Esteem Scale. To analyze the data, bivariate comparisons of quantitative variables were performed using independent samples t-tests and one-way ANOVA. Additionally, hierarchical regression analysis was conducted on happiness using two distinct sets of independent variables.

### Results

According to the results, there was a statistically significant association between marital status (p-value = 0.021), income (p-value < 0.001), education (p-value < 0.001), and physical activity (p-value < 0.001) with happiness. Happiness showed strong positive correlations with self-efficacy (r = 0.747; p-value < 0.001) and self-esteem (r = 0.306; p-value < 0.001). Hierarchical multiple linear regression analysis revealed that demographic factors accounted for 15.3% of the variance in happiness, while self-efficacy and self-esteem explained an additional 43.0%, totaling 58.7%. Among the predictors, self-efficacy was the strongest (β = 0.695).

**Data availability statement:** All relevant data are within the manuscript and its Supporting Information files.

**Funding:** The author(s) received no specific funding for this work.

## Conclusions

Self-efficacy and self-esteem were key determinants of happiness in elderly people. Healthcare centers serving older populations could implement targeted interventions such as skill-building workshops aimed at enhancing self-efficacy, cognitive-behavioral therapy (CBT) sessions to build coping strategies, and art therapy programs to boost self-esteem. Additional interventions might include mindfulness and relaxation techniques to reduce stress, social engagement activities to promote a sense of belonging, and group exercises or physical activity programs that foster both physical well-being and psychological resilience. Future studies should explore the long-term effectiveness of these interventions in promoting sustained happiness in older adults.

## Introduction

Nowadays, advancements in medical care and public health systems, including improvements in preventive medicine, early diagnosis, and treatment of chronic diseases such as cardiovascular conditions and diabetes, along with technological innovations like telemedicine and wearable health monitors, have resulted in the control and treatment of diseases [1]. Additionally, social and economic progress, the end of extensive wars, improvements in living conditions, and revolutions in agriculture have significantly contributed to increasing the human lifespan [2]. According to statistics reported in 2020, 13.5 percent of the world's population was 60 years old or older and it is estimated that by 2050, one in every five people will be older adults [3]. This demographic transition poses significant challenges for healthcare systems, which must adapt to meet the needs of older adults. They need to find effective strategies to extend health care and to respond to the needs of older adults [4].One of the health problems that are reported more frequently with age is mental health issues [5]. According to Levinson and Kaplan, self-rated mental health is more important than self-rated physical health in predicting overall self-rated health [6].

Happiness is one of the determinants of mental health [7]. Happiness is considered a necessary foundation for physical health, dealing with mental illnesses, creating emotional well-being, and increasing lifespan [6–8]. In addition, it is correlated with a stronger immune system, lower stress and depression levels, more restful sleep, enhanced cardiovascular function, greater adaptability to life events, elevated quality of life, and life satisfaction [8]. In high-income countries, happiness is an important indicator of successful aging [9]. In recent years, researchers have paid attention to happiness as a factor in promoting mental health and an important index of human mental well-being [8,10].

Various factors influence levels of happiness. Age is one of the important factors affecting happiness [11]. Studies have indicated that happiness levels decrease with increasing age [11,12]. Chronic diseases in older adults can cause physical and mental limitations and reduce their ability to perform daily activities, which can reduce their happiness [13]. When older adults have difficulty performing their daily activities due to age-related changes or become dependent on others, their belief in themselves to successfully perform a behavior changes [14]. Happiness, as a positive internal feeling, arises from a person's cognitive and emotional perception of their life. The emotional component is related to pleasure and the cognitive component is related to mental health [4]. Decreased self-efficacy in older people, as a cognitive factor, is associated with increased depression and psychological problems and decreased quality of life [15]. General self-efficacy can help older adults cope with daily struggles and stressful life events and reduce the psychological burden caused by daily problems [16].

Therefore, it can play an important role in reducing psychological problems and promoting happiness levels directly and through various mediators [17].

According to the literature, older adults with higher self-esteem are more likely to experience more positive emotions, low psychological problems, and higher levels of life satisfaction and quality of life [18,19]. The self-esteem of old people, as one of the basic needs of each person, is the extent of their understanding and appreciation of themselves and is considered a protective factor for health [19]. Seniors with higher self-esteem usually have more motivation to live and take better care of their health, which increases their life satisfaction and quality of life [20].

According to the mentioned introduction, happiness plays an important role in improving the quality of life and well-being of older adults. However, according to the 2023 World Happiness Report, Iranian seniors are ranked 103rd in happiness globally, a fact that can be attributed to specific cultural and socioeconomic factors. For instance, economic instability, limited social support networks, and cultural perceptions of aging in Iran may negatively impact the well-being and mental health of older adults [21,22]. Therefore, it is important to understand the role of cognitive factors such as self-efficacy and self-esteem and their predictive levels for happiness. This can provide valuable insights for developing effective interventions and programs to improve their quality of life and overall well-being.

## Methods

### Study design

This cross-sectional study took place in Sarab, Iran, between April to June 2023. The participants were people aged 60 years or older referring to Sarab health centers. The study received ethical approval from the Sarab University of Medical Sciences Ethics Committee with approval code IR.SARAB.REC.1402.004. In this study, the independent variables included demographic characteristics(age, gender, marital status, education and etc) as well as psychological factors like self-efficacy and self-esteem. The dependent variable was happiness, which was measured to assess its association with the aforementioned independent variables.

### Participants selection

The sample size was calculated using the Cochran formula, and the formula used was:

$$n = \frac{NZ^2P(1-P)}{d^2(N-1) + Z^2P(1-P)}$$

n = Sample size.
N = Total population size (103,324).
Z = Z-score corresponding to the desired confidence level (1.96 for 95% confidence).
P = Estimated proportion of the population (0.5).
d = Margin of error (not explicitly mentioned but assumed for calculation)

With a confidence level of 95%, power of 80% the calculated sample size was 383 participants. inclusion criteria for this study were individuals aged 60 years or older, free from psychiatric disorders, and willing to sign a consent form to participate. The only exclusion criterion was unwillingness to take part in the study.

In this study, a multi-stage cluster sampling method was employed to ensure a representative sample of the elderly population in Sarab city. In the first stage, health centers in Sarab city were divided into four clusters based on geographical locations and population density. A number of these clusters were then randomly selected to ensure a diverse representation

of the community. In the subsequent stage, a random sample of individuals aged 60 years or older was extracted from the health records of the selected clusters. This approach was chosen due to its cost-effectiveness and ease of implementation, allowing for broader coverage of the target population. By utilizing this sampling method, we aimed to minimize selection bias and enhance the generalizability of the study findings to the elderly population in Sarab city.

## Data gathering process

Data collection was conducted through face-to-face interviews with participants after explaining the study's objectives and obtaining informed consent. The interviews were carried out by two trained students at the participants' doorsteps and lasted approximately 15 to 20 minutes.

The interviewers underwent comprehensive training to ensure the consistency and reliability of data collection. This training covered the study objectives, interview techniques, and the proper use of data collection instruments. Special attention was given to minimizing interviewer bias by standardizing the questions and maintaining neutrality throughout the process.

Special attention was given to maintaining neutrality, with interviewers instructed to use neutral phrasing and avoid leading questions. Additionally, efforts were made to conduct blind interviews, where possible, to further reduce any potential biases stemming from the interviewers' expectations or assumptions about the participants. To ensure the accuracy of the data and minimize bias, interviewers were instructed to adhere to a standardized protocol and verify the completeness of responses. Regular supervision and periodic checks were implemented to monitor the quality of data collection. These measures collectively contributed to enhancing the reliability of the study's findings.

## Data collection instruments

**Demographic information.**  Demographic information includes participants' gender, marriage, education level, income status, history of smoking, physical activity, and having a disease.

**Happiness.**  The Oxford Happiness Questionnaire was used to assess happiness. Argyle and Hills expanded the Likert scale for responses from 4 to 6 points, incorporated reverse scoring for 12 items with inverted phrasing, and ultimately developed the Oxford Happiness Questionnaire to improve the accuracy of the earlier version, the Oxford Happiness Inventory. The questionnaire consists of 29 items covering seven domains: self-concept, aesthetic feelings, self-efficacy, emotional readiness, life satisfaction, hopefulness, and spiritual intelligence. Each individual can receive a score ranging from 29 to 174, which is divided by 29 to determine the overall happiness score; higher scores indicate greater happiness. Hills and Argyle reported a Cronbach's α of 0.91 for the questionnaire [23]. Additionally,Najafi et al. validated the reliability and validity of the Persian version of the questionnaire, with a Cronbach's α of 0.901 [24].

**General self-efficacy.**  To evaluate general self-efficacy, the investigators employed Sherer's questionnaire, comprising 17 statements [25]. Each statement was assessed using a 5-point Likert scale, ranging from 1 = strongly disagree to 5 = strongly agree, with a neutral option in the center. The cumulative score achievable on the questionnaire ranged from 17 to 85, with higher scores reflecting greater self-efficacy levels. The original psychometric study demonstrated high internal consistency reliability, with Cronbach's α reported as 0.86. The General Self-efficacy Questionnaire underwent validation in Iran by Asgharnezhad et al., who disclosed an Cronbach's α alpha value of 0.82, denoting satisfactory internal consistency [26].

**Rosenberg self-esteem scale (RSS).** The RSS scale was used to assess self-esteem. This scale consists of 10 statements to which participants respond by selecting one of four possible options on a 4-point Likert scale, ranging from 1 (strongly agree) to 4 (strongly disagree). The total scores on the scale range from 10 to 40 points. The original version of the scale demonstrated a Cronbach's α of 0.82, indicating good validity [27]. We used the Persian version of this questionnaire, where items 1, 2, 4, 6, and 7 require score reversal. A higher final score indicates higher self-esteem. The validity and reliability of the Persian version of the scale were confirmed by Joushanloo and Ghaedi, with a Cronbach's α of 0.83 [28].

## Data analysis

Based on the nature of the data distribution, percentages, and frequencies were utilized to analyze categorical variables, whereas metrics such as mean, standard deviation, median, and quartile deviation were employed for continuous variables. Bivariate comparisons of quantitative variables were conducted using the independent samples t-test and one-way ANOVA. The association between self-efficacy, self-esteem, and happiness was assessed through the utilization of Pearson correlation analysis.

Two distinct sets of independent variables were utilized in conducting a hierarchical regression analysis on happiness. Hierarchical regression was chosen because it allows for assessing the incremental contribution of additional predictors in explaining the variance in happiness. In the initial block (Block 1), demographic variables were included, as these foundational factors often influence psychological outcomes. Subsequently, in Block 2, self-efficacy and self-esteem were added to examine whether these psychological factors contribute significantly to the explanation of happiness beyond demographic characteristics. The proportion of variance associated with happiness was determined by assessing the adjusted $R^2$ change following the introduction of each block. The validity of the regression assumptions was confirmed through tests for multicollinearity, normality, and statistical significance. A significance threshold of $\alpha = 0.05$ was established. All statistical analyses were carried out using SPSS version 26.

## Ethics statement

The Ethics Committee of the Sarab Faculty of Medical Sciences (Ethics Code: IR.SARAB. REC.1402.004) approved this research. Written consent was obtained from the participants.

## Results

In the study, a total of 400 elders agreed to participate. Among the subjects, 51% were male, and most of them were married (71.3%) and illiterate (43.0%). Since the data collection was conducted through face-to-face interviews, all data were comprehensively collected, and no missing data were encountered. According to the findings, there was a statistically significant association between marital status (p-value = 0.021), status of income (p-value < 0.001), education (p-value < 0.001), and physical activity (p-value < 0.001) with happiness. Table 1 shows the demographic characteristics and their relationship to happiness.

The bivariate associations for self-efficacy and self-esteem with happiness are displayed in Table 2. Using the Pearson correlation coefficient test, we discovered that happiness had a statistically significant correlation with self-efficacy (r = 0.747; p-value < 0.001) and self-esteem (r = 0.306; p-value < 0.001).

To predict happiness, we used hierarchical multiple linear regression. In step 1, demographic variables were significant predictors of happiness (p-value < 0.001, $R^2 = 0.153$),

**Table 1. Demographic characteristics of the elders and their association with happiness.**

| Variable | | N (%) | Happiness | p-value* |
|---|---|---|---|---|
| | | | Me ± SD | |
| Gender ** | Male | 204 (0) | 88.06 ± 17.95 | 0.428 |
| | Female | 196 (29) | 86.58 ± 19.52 | |
| Marriage ** | Married | 285 (71.3) | 88.71 ± 18.90 | 0.021 |
| | Single | 115 (28.8) | 83.93 ± 17.93 | |
| status of income *** | Pension | 57 | 95.85 ± 21.23 | 0.001 |
| | Saving | 51 | 85.16 ± 19.91 | |
| | Depend on spouse | 101 | 86.78 ± 18.25 | |
| | Depend on child | 144 | 83.95 ± 17.18 | |
| | self-employment | 47 | 90.91 ± 16.61 | |
| Education** | Under diploma | 299 | 85.28 ± 16.59 | 0.001 |
| | Diploma and higher | 101 | 93.43 ± 23.01 | |
| History of smoking** | No | 327 | 87.39 ± 18.93 | 0.902 |
| | Yes | 73 | 87.09 ± 17.90 | |
| Physical activity ** | No | 370 | 85.91 ± 17.71 | 0.001 |
| | Yes | 30 | 104.86 ± 21.58 | |
| Having diseases ** | No | 124 | 89.79 ± 19.88 | 0.079 |
| | Yes | 276 | 86.23 ± 18.12 | |

*P-value < 0.05,

**Independent sample T-test,

***ONE Way ANOVA Test.

**Table 2. Bivariate correlation matrix of the relationship between Happiness, self-efficacy, and self-esteem.**

| Variables | 1 | 2 | 3 | Me ± SD |
|---|---|---|---|---|
| 1 = Self-efficacy | 1 | | | 12.31 ± 7.92 |
| 2 = Self-esteem | 0.354* | 1 | | 18.48 ± 8.48 |
| 3 = Happiness | 0.747* | 0.306* | 1 | 22.78 ± 8.22 |

*Correlation is significant at the 0.05 level (two-tailed).

explaining 15.3% of the variation, as shown in Table 3. This suggests that demographic factors alone account for a limited but notable portion of the variability in happiness (F = 8.83; p-value < 0.001). In step 2, adding self-efficacy and self-esteem substantially improved the model, increasing the R² to 0.587 and explaining an additional 43.0% of the variation (F = 204.02; p-value < 0.001). This indicates that self-efficacy and self-esteem significantly contribute to predicting happiness beyond demographic variables. Self-efficacy ($\beta = 0.695$) emerged as the most impactful predictor. The results highlight that while demographic factors are foundational, psychological attributes such as self-efficacy and self-esteem play a predominant role in shaping happiness among elders.

The findings of this study highlight self-efficacy as the most significant predictor of happiness among the elderly population. This underscores the pivotal role of an individual's belief in their capability to manage life's challenges and maintain autonomy, which are critical components of psychological well-being in older age. Additionally, self-esteem emerged as a significant contributor, reflecting the importance of a positive self-view in fostering emotional resilience and facilitating social relationships.

**Table 3. Hierarchical linear regression for adherence to medication through demographic characteristics, HL dimensions, and illness perception.**

| Variables | ß | R2 change | F change | SE | P-value* |
|---|---|---|---|---|---|
| **Step 1** | | | | | |
| Age | -0.263* | 0.153 | 8.83 | 0.130 | 0.001 |
| Gender | 0.015 | | | 1.860 | |
| Marriage | 0.017 | | | 2.046 | |
| Income | -0.061 | | | 0.636 | |
| Education level | 0.068 | | | 2.170 | |
| Smoking | 0.020 | | | 2.403 | |
| Physical activity | 0.195* | | | 3.463 | |
| Having chronic disease | 0.021 | | | 1.984 | |
| **Step 2** | | | | | |
| Age | -0.110 | 0.43 | 204.02 | 0.093 | < 0.001 |
| Gender | 0.020 | | | 1.305 | |
| Marriage | 0.043 | | | 1.446 | |
| Income | -0.013 | | | 0.447 | |
| Education level | 0.006 | | | 1.536 | |
| Smoking | -0.049 | | | 1.684 | |
| Physical activity | 0.098 | | | 2.458 | |
| Having chronic disease | 0.022 | | | 1.396 | |
| **Self-esteem** | 0.695* | | | 0.065 | |
| **Self-efficacy** | 0.037 | | | 0.152 | |
| Total R2 | – | 0.587 | – | – | – |
| Adjusted R2 | – | 0.577 | – | – | – |

*P-value < 0.05.

## Discussion

Mental health and illnesses are more frequently with increasing age. Happiness is a key determinant of mental well-being in the elderly who face various health problems and their complications. This research aimed to investigate the association of self-efficacy and self-esteem with happiness in elders. According to the findings, there was a significant positive relationship between, being married, high level of education, being retired and having physical activity with happiness. with happiness. This result is in line with the other studies about marital status, type of income, and physical activity [29]. According to the results of research in Turkey, married people were happier and education level had a positive effect on happiness [30]. A similar finding was found in a study done in Abu Dhabi that showed marital status, and education attainment were factors affecting happiness in older people [31]. The founding of study in Hamadan, Iran, indicated there was a significant relationship between happiness and some socio-demographic characteristics such as educational level and income status [8]. Results of the study showed social support moderate the indirect relationship between self-efficacy and happiness through mental health in the elderly [17]. Also, social support by significant others is associated with psychological well-being [32]. High education levels and independence in income help to increase self-esteem and self-efficacy in these elderly people. The results of a study conducted by Toshkov, reported income had a main role in happiness in a large number of European countries [33]. A systematic review of the literature study conducted among older people in 2024 showed a similar result, i.e., that economic, social and health-related factors were associated with happiness. The sociodemographic characteristics

of older people including the female sex, being married, an older age, a higher level of schooling and having a religion, were determinants of happiness [34]. In line with the results of our study, the findings of the research in South Korea demonstrated marital status, age, period of residence, income level, area of residence (urban or rural), religion, occupation type, residence type, gender and education level were predictors of happiness of local residents [35]. Therefore, the happiness in older people differs by socioeconomic status and more support from policymakers is needed. The design of intervention programs by health care providers cannot be successful without considering these factors. Furthermore, having physical activity in older people can prevent social isolation and unhappiness. Having regular physical exercise is effective in improving the quality of life among older adults [36]. The results of a study conducted in China indicated physical exercise had a positive effect on subjective well-being levels among older adults [37]. Moreover, self-esteem mediated the relationship between physical exercise and the subjective well-being of older adults [37]. A positive correlation was reported between life satisfaction, self-esteem, and self-efficacy among regular old adult exercisers [36]. Physical activity programs are effective in mental health, for the successful aging and self-esteem of elderly women [38]. It is necessary to design community–based educational and practical programs to increase physical activity such as holding morning exercise in parks in the older people by municipality. The results of study in Tehran, Iran, demonstrated self-efficacy had a positive relationship with psychological well-being and also with quality of life [15].

The bivariate associations for self-efficacy and self-esteem with happiness displayed that happiness had a statistically significant correlation with self-efficacy and self-esteem. The findings suggest that raising self-esteem, and self-efficacy can enhance happiness in elderly people. For instance, potential mechanisms can be improved coping strategies, greater social participation, or reduced psychological distress. The results of a study conducted in Thailand revealed that self-esteem was a predictor of happiness among older adults [39]. In a study, Erozkan reported a significant impact of persistence sub-dimensions of self-efficacy and internal self-confidence and external self-confidence sub-dimensions of self-esteem on subjective happiness [39]. Also, the other previous study demonstrated emotional self-efficacy had a positive effect on self-esteem and self-esteem had a positive effect on happiness [40]. The results of the recent study were consistent with the results of the study conducted in 2020 which indicated a relationship between self-esteem and happiness [41]. In older adults, there is a relationship between healthy lifestyle, self-esteem, and subjective vitality [42]. Considering this fact, elderly people with high self-efficacy, and high self-esteem are more likely to be happy and have a healthy lifestyle. Focusing on this issue most elderly people suffer from various health problems and diseases and they have to adhere to special diet and medication [43,44]. which may not be pleasant. Hence, it is important to design intervention programs in terms of improving self-efficacy, self-esteem, and happiness among elderly people in health centers and happiness competitions through Mass Media/ TV.

According to the founding of hierarchical multiple linear regression, demographic characteristics explained 15.3 percent of the variation in happiness. Step 2 included the addition of self-efficacy and self-esteem, which explained an additional 43.0% of the variation. In total, demographic factors, self-efficacy, and self-esteem were able to account for 58.7% of the variation in happiness. Self-efficacy was the most important predictor of happiness among the variables. When an elder is self-efficacious, his or her self-esteem seems more likely to be improved and translated into raised happiness. This relation seems to highlight the probable positive outcomes of increasing self-efficacy on self-esteem and happiness of elderly adults and therefore on the quality of life. Self-efficacy is an important factor in implementing health behaviors in elder adults [45]. A study conducted in Iran (2023) showed spiritual health, social support, and socio-demographic variables explained 43% of the variance of change in

happiness [30]. The other study in Iran implied spirituality and self-efficacy predicted 29% and 36% of happiness variance, respectively [46].

Founding of a study in Northeast Thailand revealed happiness of elderly relationship with family living style, average monthly income, relationships with family, access to human capital, and the livelihoods. These factors explained the happiness of the elderly 36.7% ($R^2$ = 0.367) [47].

Our findings are consistent with the previous study, which analyzed the relationship between self-esteem and happiness, the proportion of variance for happiness as explained by the model was $R^2 = 0.64$ [41]. Similarly, meaning self-esteem of older adults and preparation for aging were determinants of happiness, accounting for 67.0% of the variance of happiness [48]. The other research reported that psychological well-being, emotional self-efficacy, affect balance and self-esteem explained 51% of the total variance regarding happiness [40]. Most of the programs for elderly people focus on improving physical health, but it is time to enhance psychology-based programs in healthcare service centers and communities. It is important to consider socio-demographic features in the design of interventions to increase happiness in elderly people, especially a certain subgroup of the elderly population. In the meantime, lifelong learning initiatives, support groups, and art therapy interventions can be useful in improving self-efficacy and self-esteem, and subsequently increasing happiness in the older people.

## Limitations

There were some limitations in this study. The self-reporting measures used in the study might be biased in responders. This cross-sectional study focused on the association between individual factors and happiness among older people, which may have ignored the importance of family and community factors such as social support. Also, this research underrepresents a certain subgroup of the elderly population in Sarab. A study was conducted in a city located in the Northwest of Iran; thus, the results need to be generalized more carefully outside the Iranian context, because cultural differences might affect self-efficacy, self-esteem, and happiness relationships.

## Conclusions

In general, being retired, having a high level of education, marital status, and physical activity are highly positively associated with happiness in older adults. Regarding psychological resources, happiness was also positively and statistically significantly related to self-efficacy and self-esteem. Demographic factors, self-efficacy, and self-esteem together accounted for 58.7% of the total variance in happiness, whereby self-efficacy emerged as the strongest predictor. In the light of these findings, health interventions in the pursuit of happiness among older adults are better tailored to focus on both demographic variables and psychological factors. Concretely, health providers and policymakers should also consider targeted programs such as cognitive-behavioral therapy sessions that enhance self-efficacy and emotional regulation. This would further be reinforced if skills-building workshops allowing the older adult to focus on, and achieve, personal goals were included. These can significantly build their self-esteem and well-being. Other interventions may involve mindfulness and relaxation to decrease stress, social engagement to improve a sense of belonging, group exercises or physical activity programs that promote not only physical but also psychological resilience. These actionable suggestions may act as a building block in constructing detailed guidelines that could enhance happiness among Iranian older adults. Prioritizing intervention programs for the enhancement of self-efficacy and self-esteem should continue with the objective of better mental health and higher quality of life among this demographic. Future research needs

to consider to what extent proposed interventions might lead to long-lasting effects on the promotion of happiness among older adults.

## Supporting information

**S1 Table. Demographic characteristics of the elders and their association with happiness.**
(PDF)

**S2 Table. Bivariate correlation matrix of the relationship between Happiness, self-efficacy, and self-esteem.**
(DOCX)

**S3 Table. Hierarchical linear regression for adherence to medication through demographic characteristics, HL dimensions, and illness perception.**
(PDF)

**S1 File. Study Dataset.**
(XLSX)

**S2 File. Approval Document.**
(DOCX)

## Acknowledgments

The authors acknowledge the support of the Sarab Faculty of Medical Sciences and thank all elderly people and staff of the Healthcare Service Centers (HSCs) for providing valuable data.

## Author contributions

**Conceptualization:** Soheila Ranjbaran, Khalil Maleki Chollou.

**Data curation:** Sara Pourrazavi, Erfan Saeedi Tazekand.

**Formal analysis:** Towhid Babazadeh, Sara Pourrazavi.

**Investigation:** Towhid Babazadeh, Soheila Ranjbaran, Khalil Maleki Chollou, Erfan Saeedi Tazekand.

**Methodology:** Towhid Babazadeh, Sara Pourrazavi, Khalil Maleki Chollou.

**Resources:** Akbar Nadi.

**Software:** Towhid Babazadeh.

**Supervision:** Khalil Maleki Chollou, Akbar Nadi.

**Validation:** Towhid Babazadeh, Soheila Ranjbaran.

**Visualization:** Sara Pourrazavi.

**Writing – original draft:** Towhid Babazadeh, Khalil Maleki Chollou.

**Writing – review & editing:** Towhid Babazadeh, Soheila Ranjbaran, Sara Pourrazavi, Khalil Maleki Chollou, Akbar Nadi, Erfan Saeedi Tazekand.

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
