## [Decision Letter · Decision Letter 0]

24 Jun 2024

PONE-D-24-19287Association self-efficacy and self-esteem with Happiness in elders: A cross-sectional studyPLOS ONE

Dear Dr. Maleki Chollou,

Thank you for submitting your manuscript to PLOS ONE. After careful consideration, we feel that it has merit but does not fully meet PLOS ONE’s publication criteria as it currently stands. Therefore, we invite you to submit a revised version of the manuscript that addresses the points raised during the review process.

<h3>Refined Research Title Suggestions:</h3>

**Happiness in the Elderly: Self-Efficacy and Self-Esteem in a Cross-Sectional Study****Self-Efficacy, Self-Esteem, and Happiness in Older Adults: A Cross-Sectional Study****Linking Self-Efficacy, Self-Esteem, and Happiness in Elders: A Cross-Sectional Study**

<h3>Abstract:</h3>

**Background:** Briefly outline theoretical or empirical foundations for studying self-efficacy and self-esteem in relation to elderly happiness.**Measurement Tools:** Specify the scales or tools used to assess self-efficacy, self-esteem, and happiness.**Conclusion:** Offer specific suggestions for programs or interventions to enhance self-efficacy and self-esteem among the elderly.

<h3>Introduction:</h3>

**Specificity:** Detail specific advancements in medical care, public health, and socio-economic progress.**Redundancy:** Combine life expectancy increase statements for conciseness.**Transition:** Smooth the transition from demographic statistics to health priorities.**Explanation:** Provide context or examples for the assertion that mental health is crucial for older adults.**Citation:** Place citations immediately after relevant statistics for clarity.

<h3>Methods:</h3>

**Sampling Formula:** Clearly format and explain terms in the sample size formula.**Criteria Justification:** Explain the choice of hypertension as an inclusion criterion and its relation to the study's focus.**Sampling Method:** Clarify the term "multi-state cluster random method" and describe participant selection.**Data Collection:** Elaborate on the data collection process, interviewer training, and steps to ensure data accuracy and minimize bias.**Instrument Validation:** Detail the validation process for instruments in the local context.**Missing Data:** Describe how missing data were handled.**Ethical Considerations:** Mention ethical approval and adherence to ethical standards.**Software Version:** Update or confirm the statistical software version used.**Hierarchical Regression:** Provide a detailed rationale for choosing hierarchical regression and block determination.

<h3>Discussion:</h3>

**Streamlining:** Remove redundancies, particularly regarding the association of self-efficacy and self-esteem with happiness.**Depth:** Deepen the analysis by exploring underlying reasons for results.**Term Consistency:** Define and consistently use terms like self-confidence.**Correlations and Causation:** Discuss potential causation and other influencing factors beyond correlations.**Practical Implications:** Offer specific recommendations for interventions.**Transitions:** Improve transitions between points for better readability.**Comparison with Literature:** Compare findings with existing literature and address study limitations, including cross-sectional design and generalizability.**Statistical Results:** Avoid unnecessary repetition of statistical results.

We look forward to receiving your revised manuscript.

Kind regards,

Ayman Mohamed El-Ashry, Associate professor, PhD, MD,BNs, ACT

Academic Editor

PLOS ONE

Journal Requirements:

2. In the online submission form, you indicated that The data used for this research is available on request from the corresponding author.

Additional Editor Comments:

Refined Research Title Suggestions:

Happiness in the Elderly: Self-Efficacy and Self-Esteem in a Cross-Sectional Study

Self-Efficacy, Self-Esteem, and Happiness in Older Adults: A Cross-Sectional Study

Linking Self-Efficacy, Self-Esteem, and Happiness in Elders: A Cross-Sectional Study

Abstract:

Background: Briefly outline theoretical or empirical foundations for studying self-efficacy and self-esteem in relation to elderly happiness.

Measurement Tools: Specify the scales or tools used to assess self-efficacy, self-esteem, and happiness.

Conclusion: Offer specific suggestions for programs or interventions to enhance self-efficacy and self-esteem among the elderly.

Introduction:

Specificity: Detail specific advancements in medical care, public health, and socio-economic progress.

Redundancy: Combine life expectancy increase statements for conciseness.

Transition: Smooth the transition from demographic statistics to health priorities.

Explanation: Provide context or examples for the assertion that mental health is crucial for older adults.

Citation: Place citations immediately after relevant statistics for clarity.

Methods:

Sampling Formula: Clearly format and explain terms in the sample size formula.

Criteria Justification: Explain the choice of hypertension as an inclusion criterion and its relation to the study's focus.

Sampling Method: Clarify the term "multi-state cluster random method" and describe participant selection.

Data Collection: Elaborate on the data collection process, interviewer training, and steps to ensure data accuracy and minimize bias.

Instrument Validation: Detail the validation process for instruments in the local context.

Missing Data: Describe how missing data were handled.

Ethical Considerations: Mention ethical approval and adherence to ethical standards.

Software Version: Update or confirm the statistical software version used.

Hierarchical Regression: Provide a detailed rationale for choosing hierarchical regression and block determination.

Discussion:

Streamlining: Remove redundancies, particularly regarding the association of self-efficacy and self-esteem with happiness.

Depth: Deepen the analysis by exploring underlying reasons for results.

Term Consistency: Define and consistently use terms like self-confidence.

Correlations and Causation: Discuss potential causation and other influencing factors beyond correlations.

Practical Implications: Offer specific recommendations for interventions.

Transitions: Improve transitions between points for better readability.

Comparison with Literature: Compare findings with existing literature and address study limitations, including cross-sectional design and generalizability.

Statistical Results: Avoid unnecessary repetition of statistical results.

Reviewers' comments:

Reviewer's Responses to Questions

**Comments to the Author**

1. Is the manuscript technically sound, and do the data support the conclusions?

Reviewer #1: Yes

Reviewer #2: Yes

2. Has the statistical analysis been performed appropriately and rigorously? 

Reviewer #1: Yes

Reviewer #2: Yes

3. Have the authors made all data underlying the findings in their manuscript fully available?

Reviewer #1: Yes

Reviewer #2: Yes

4. Is the manuscript presented in an intelligible fashion and written in standard English?

Reviewer #1: Yes

Reviewer #2: No

5. Review Comments to the Author

**Reviewer #1: ** Title

Your research title is clear and informative, but it can be made more concise and engaging. Here are a few suggestions to refine and enhance the title:

• Happiness in the Elderly: The Role of Self-Efficacy and Self-Esteem in a Cross-Sectional Study

• The Relationship Between Self-Efficacy, Self-Esteem, and Happiness in Older Adults: A Cross-Sectional study

• Exploring the Link Between Self-Efficacy, Self-Esteem, and Happiness Among Elders: A Cross-Sectional Study

Abstract

• Expand the background to briefly explain the theoretical or empirical basis for focusing on self-efficacy and self-esteem in relation to happiness in the elderly.

• Mention the specific tools or scales used to measure self-efficacy, self-esteem, and happiness.

• Conclusion: The conclusion is somewhat vague regarding how to design programs to increase self-efficacy and self-esteem. Suggest specific types of programs or interventions that could be implemented to enhance self-efficacy and self-esteem among the elderly.

Introduction

• Lack of Specificity: The phrase "advancements in medical care, public health, and socio-economic progress" is vague. Providing specific examples or types of advancements would enhance clarity

• Redundancy: The idea that life expectancy has increased leading to a larger elderly population is repeated in the statistics that follow. Combining these statements could make the paragraph more concise.

• Abrupt Transition: The transition from demographic statistics to the discussion on shifting health priorities is abrupt. A connecting sentence would improve the flow.

• Insufficient Explanation: The claim that "mental health is even more crucial than physical health for older adults" is a significant assertion that needs more context or examples to substantiate why mental health is deemed more critical.

• Citation Placement: The placement of citations could be more precise. For example, placing the citation immediately after the relevant statistics (e.g., after "In 2020, 13.5% of the global population was 60 years or older (2).

Methods

• Clarity in Sampling Formula: The sample size formula is presented as a mathematical equation but isn't clearly formatted for readability. Additionally, there is no explanation of the terms used in the formula.

• Inclusion and Exclusion Criteria: The inclusion criteria mention "diagnosed with hypertension," but it's unclear why this specific condition was chosen or how it relates to the study's focus on happiness, self-efficacy, and self-esteem.

• Sampling Method Description: The term "multi-state cluster random method" is used, which might be confusing. It should be clarified if this means multi-stage cluster sampling and how exactly the clusters and participants were selected.

• Data Collection Process: The description of the data collection process is brief and lacks details on how the interviews were conducted, how data accuracy was ensured, and if any steps were taken to minimize bias.

• Interviewer Training: There is no mention of how the two educated students who conducted the interviews were trained, which could affect the consistency and reliability of data collection.

• Validation of Instruments: While the internal consistency of the instruments is mentioned, more information on the process of validation, especially in the context of the local population, would strengthen the methodology.

• Handling Missing Data: There is no mention of how missing data were handled, which is important for ensuring the validity of the results.

• Ethical Considerations: There is no explicit mention of ethical approval or the process followed to ensure ethical standards were maintained during the study.

• Statistical Software Version: The mention of "SPSS 21 software" could be updated to ensure readers know the analysis was done using a relevant and recent version of the software.

• Details on Hierarchical Regression: The explanation of hierarchical regression could be more detailed, particularly on why this method was chosen and how the blocks were determined.

Discussion

The discussion section has several weak points that need addressing. Redundancies, such as repeating the association of self-efficacy and self-esteem with happiness, should be streamlined for clarity. The analysis lacks depth, providing a summary of results without a deeper exploration of underlying reasons. Terms like self-confidence are inconsistently used and not clearly defined, potentially confusing readers. There is also an over-reliance on correlations without discussing potential causation or other influencing factors. Practical implications are vague, with no specific recommendations for interventions. Transitions between points are abrupt, making the discussion difficult to follow. Additionally, the discussion does not sufficiently compare findings with existing literature or address study limitations, such as the cross-sectional design and generalizability. Finally, some statistical results are repeated without adding new insights, unnecessarily lengthening the discussion.

Please address implications for nursing research, education and practice

**Reviewer #2: ** This article is well-written and talks about an important issue that requires attention and significant consideration by health professionals. But it has some problems which I will mention below. I hope that by reforming them, the work can be improved.

- It is better to use term "older people “instead of "elderly", “elders"

- Put more information in the abstract, method section.

- In view of the high prevalence of chronic diseases such as high blood pressure during old age, how many were the total number of elderly people without high blood pressure in Sarab city, out of which 400 samples were randomly selected? How to select and access samples should also be mentioned.

- Please explain why the authors chose people without blood pressure?

-

- It is written in the working method section that the questionnaires were completed by the questioner. In this case, please explain what measures were used to avoid interfering with the questioner's opinion in completing the information?

- In the demographic information section, what type of disease did you mean by disease?

- It is better to mention having or not having a disease instead of writing the disease.

- Regarding the happiness tool, the mentioned reference is related to the research done in India, please put the reference related to the Oxford Happiness Questionnaire first, and then the validity and reliability of the questionnaire in Persian and English version, along with mentioning the references to be mentioned separately. Also, write the minimum and maximum score of this questionnaire. If the validity and reliability of this questionnaire has been done in your article, please mention it. Also explain how to interpret the score obtained from the questionnaire.

- In relation to the self-efficacy questionnaire, the reference mentioned does not match what was referred. Please refer to the original author of the questionnaire. Also, Cronbach's alpha should be written both in the main study and in the study conducted in Iran. Cronbach's alpha should be written instead of the term alpha value.

- Regarding the self-esteem scale questionnaire, please add the original reference and write its psychometric description.

- Please specify the independent and dependent variables of this research.

- In TABLE 1 regarding the marital status, only two groups, single and married, are mentioned, while there are also divorced people and deceased spouses. What has been done about these people?

- Instead of income, write the status of income.

- Why is education divided into two categories below diploma and above diploma? If it seems that the classification of elderly people in the sub-diploma category with more classes is more important. Because most of the elderly do not have a high level of education. Also, did you have an illiterate person in this study? This is despite the fact that you mentioned in the findings section that 43% of people were illiterate.

- In the discussion section, the statistically significant type should be mentioned and explained. You are the only one who mentions that there is a statistically significant association. You should write, for example, with increasing physical activity, the level of happiness increased and...

6. PLOS authors have the option to publish the peer review history of their article (what does this mean? ). If published, this will include your full peer review and any attached files.

**Do you want your identity to be public for this peer review?** For information about this choice, including consent withdrawal, please see our Privacy Policy .

Reviewer #1: **Yes: ** shaimaa mohamed amin

Reviewer #2: No

---

## [Author Response · Author response to Decision Letter 0]

9 Sep 2024

Title: Response to Reviewer Comments on Manuscript

To the Editor of Plos One : Dr Ayman Mohamed El-Ashry

Manuscript : PONE-D-24-19287

Dear Editor,

On behalf of my co-authors, I, Khalil Maleki Chollou, would like to express our sincere gratitude to you and the reviewers for the thorough review and valuable feedback on our manuscript . Your comments will greatly help us improve the quality of our paper.

In this letter, we intend to address each of the reviewers' comments and suggestions in detail.

Suggestion, Question,

or Comment from the

Editor Author’s Response Change in the Manuscript

Refined Research Title Suggestions:

Happiness in the Elderly: Self-Efficacy and Self-Esteem in a Cross-Sectional Study

Self-Efficacy, Self-Esteem, and Happiness in Older Adults: A Cross-Sectional Study

Linking Self-Efficacy, Self-Esteem, and Happiness in Elders: A Cross-Sectional Study

We truly appreciate the Editor’s suggestion, and we recognize that it would improve the revised version of the manuscript.

Change of Manuscript Title from “Association self-efficacy and self-esteem with Happiness in elders: A cross-sectional study” to “Linking Self-Efficacy, Self-Esteem and Happiness in Older People: A Cross-Sectional Study”

Abstract:

Background: Briefly outline theoretical or empirical foundations for studying self-efficacy and self-esteem in relation to elderly happiness.

Measurement Tools: Specify the scales or tools used to assess self-efficacy, self-esteem, and happiness.

Conclusion: Offer specific suggestions for programs or interventions to enhance self-efficacy and self-esteem among the elderly.

We have meticulously reviewed your feedback and have implemented substantial revisions to the abstract, enhancing its clarity, conciseness, and overall impact. Your valuable insights have been instrumental in refining the abstract to accurately reflect the core findings and significance of the research In line 33-35, the Background was modified with the following paragraph:

“Happiness is crucial for well-being in older people, but it can be challenged by various health issues. Self-efficacy, the belief in one's ability to manage challenges, and self-esteem, positive self-regard, are theorized to play significant roles in promoting happiness in this population.”

In line 39-45, the Methods was modified with the following paragraph:

“A cross-sectional study was conducted with 400 individuals aged 60 years or older who visited health centers in Sarab, Iran, from April to June 2023. Data were collected using valid and reliable instruments, including the Oxford Happiness Questionnaire, Sherer’s Self-Efficacy Scale, and the Rosenberg Self-Esteem Scale. To analyze the data, bivariate comparisons of quantitative variables were performed using independent samples t-tests and one-way ANOVA. Additionally, hierarchical regression analysis was conducted on happiness using two distinct sets of independent variables.”

In line 56-61, the Conclusion was modified with the following paragraph:

“To enhance happiness in elderly populations, healthcare centers frequented by this demographic could implement programs designed to bolster self-efficacy and self-esteem. Examples include lifelong learning initiatives, support groups, and art therapy interventions. The use of effectiveness of such interventions in promoting happiness among older adults can promote the happiness in elderly people warrants further investigation in future studies..”

Introduction:

Specificity: Detail specific advancements in medical care, public health, and socio-economic progress.

Redundancy: Combine life expectancy increase statements for conciseness.

Transition: Smooth the transition from demographic statistics to health priorities.

Explanation: Provide context or examples for the assertion that mental health is crucial for older adults.

Citation: Place citations immediately after relevant statistics for clarity. We have meticulously reviewed your feedback and have implemented substantial revisions to the Introduction, enhancing its clarity, conciseness, and overall impact. Your valuable insights have been instrumental in refining the Introduction to accurately reflect the core findings and significance of the research Significant revisions were made throughout the introduction to enhance clarity and conciseness.

Sampling Formula: Clearly format and explain terms in the sample size formula. Thank you for your valuable feedback. We have revised the description of the sample size calculation to provide a clearer and more detailed explanation. In line 134-141, was modified.

Criteria Justification: Explain the choice of hypertension as an inclusion criterion and its relation to the study's focus. Thank you for your insightful comment. We appreciate the opportunity to clarify this point. The study was designed to include elderly participants regardless of their hypertension status, as we aimed to capture a broad spectrum of health conditions within the elderly population. The mention of hypertension as an inclusion criterion was not intended to imply exclusivity, but rather to acknowledge the commonality of such conditions in the target demographic. In practice, the study included both individuals with and without hypertension to ensure a comprehensive analysis. We have revised the manuscript to more accurately reflect this approach.

Sampling Method: Clarify the term "multi-state cluster random method" and describe participant selection. Thank you for your comment. We have clarified the sampling method and participant selection process in the revised manuscript In line 146-151, was modified.

Data Collection: Elaborate on the data collection process, interviewer training, and steps to ensure data accuracy and minimize bias. Thank you for your comment. We have expanded on the data collection process, including interviewer training and measures taken to ensure data accuracy and minimize bias. In line 154-165, was modified.

Instrument Validation: Detail the validation process for instruments in the local context. Thank you for your comment. All necessary revisions have been made to ensure that the validation process is detailed and specific to the local context. In line 171-204, was modified.

Missing Data: Describe how missing data were handled. Thank you for your comment. As we employed a face-to-face interview technique for data collection, every effort was made to ensure that all data were collected comprehensively. Due to the direct interaction between interviewers and participants, no data were missing, and all questionnaires were fully completed. In line 239-241, was added

Ethical Considerations: Mention ethical approval and adherence to ethical standards. Thank you for your comment.Done In line 126-128, was modified

Software Version: Update or confirm the statistical software version used. Thank you for your comment. The statistical analyses were performed using SPSS version 26. We have updated the manuscript to reflect the correct version of the software used.

Hierarchical Regression: Provide a detailed rationale for choosing hierarchical regression and block determination. Thank you for your insightful comment. Hierarchical regression was chosen for this study because it allows for the assessment of the incremental value of additional predictors in explaining the variance in happiness. By entering variables in blocks, we could observe how much additional variance was explained by including psychological factors such as self-efficacy and self-esteem after accounting for basic demographic characteristics.

Block 1 included demographic variables because these factors are foundational and often influence psychological outcomes. In Block 2, self-efficacy and self-esteem were added to examine whether these psychological factors provide a significant contribution to the explanation of happiness, beyond what is explained by demographic variables alone. The change in adjusted R² between these blocks provided insight into the unique contribution of these psychological variables.

We have revised the manuscript to incorporate this rationale more explicitly and to clarify the determination of the blocks. In line 219-229, was modified

Discussion:

Streamlining: Remove redundancies, particularly regarding the association of self-efficacy and self-esteem with happiness.

Depth: Deepen the analysis by exploring underlying reasons for results.

Term Consistency: Define and consistently use terms like self-confidence.

Correlations and Causation: Discuss potential causation and other influencing factors beyond correlations.

Practical Implications: Offer specific recommendations for interventions.

Transitions: Improve transitions between points for better readability.

Comparison with Literature: Compare findings with existing literature and address study limitations, including cross-sectional design and generalizability.

Statistical Results: Avoid unnecessary repetition of statistical results. All the comments were made point-by-point and marked by track changes and highlighting in the revision file of the manuscript.

Suggestion, Question,

or Comment from

Reviewer #1 Author’s Response Change in the Manuscript

Your research title is clear and informative, but it can be made more concise and engaging. Here are a few suggestions to refine and enhance the title:

• Happiness in the Elderly: The Role of Self-Efficacy and Self-Esteem in a Cross-Sectional Study

• The Relationship Between Self-Efficacy, Self-Esteem, and Happiness in Older Adults: A Cross-Sectional study

• Exploring the Link Between Self-Efficacy, Self-Esteem, and Happiness Among Elders: A Cross-Sectional Study We truly appreciate the Reviewer #1 suggestion, and we recognize that it would improve the revised version of the manuscript.

Change of Manuscript Title from “Association self-efficacy and self-esteem with Happiness in elders: A cross-sectional study” to “Linking Self-Efficacy, Self-Esteem and Happiness in Older People: A Cross-Sectional Study”

Abstract

• Expand the background to briefly explain the theoretical or empirical basis for focusing on self-efficacy and self-esteem in relation to happiness in the elderly.

• Mention the specific tools or scales used to measure self-efficacy, self-esteem, and happiness.

• Conclusion: The conclusion is somewhat vague regarding how to design programs to increase self-efficacy and self-esteem. Suggest specific types of programs or interventions that could be implemented to enhance self-efficacy and self-esteem among the elderly. We have meticulously reviewed your feedback and have implemented substantial revisions to the abstract, enhancing its clarity, conciseness, and overall impact. Your valuable insights have been instrumental in refining the abstract to accurately reflect the core findings and significance of the research In line 33-35, the Background was modified with the following paragraph:

“Happiness is crucial for well-being in older people, but it can be challenged by various health issues. Self-efficacy, the belief in one's ability to manage challenges, and self-esteem, positive self-regard, are theorized to play significant roles in promoting happiness in this population.”

In line 39-45, the Methods was modified with the following paragraph:

“A cross-sectional study was conducted with 400 individuals aged 60 years or older who visited health centers in Sarab, Iran, from April to June 2023. Data were collected using valid and reliable instruments, including the Oxford Happiness Questionnaire, Sherer’s Self-Efficacy Scale, and the Rosenberg Self-Esteem Scale. To analyze the data, bivariate comparisons of quantitative variables were performed using independent samples t-tests and one-way ANOVA. Additionally, hierarchical regression analysis was conducted on happiness using two distinct sets of independent variables.”

In line 56-61, the Conclusion was modified with the following paragraph:

“To enhance happiness in elderly populations, healthcare centers frequented by this demographic could implement programs designed to bolster self-efficacy and self-esteem. Examples include lifelong learning initiatives, support groups, and art therapy interventions. The use of effectiveness of such interventions in promoting happiness among older adults can promote the happiness in elderly people warrants further investigation in future studies..”

Lack of Specificity: The phrase "advancements in medical care, public health, and socio-economic progress" is vague. Providing specific examples or types of advancements would enhance clarity

Redundancy: The idea that life expectancy has increased leading to a larger elderly population is repeated in the statistics that follow. Combining these statements could make the paragraph more concise.

Thank you for this valuable suggestion. In response,we have revised the manuscript to include specific examples of advancements in these areas. In line 74-77, was modified with the following paragraph:

“Nowadays, with advancements in medical care and public health system resulting in the control and treatment of diseases, as well as social and economic progress that have led to the end of extensive wars, improvements in living conditions and the revolutions in the agriculture, increased the lifespan of humans”

Abrupt Transition: The transition from demographic statistics to the discussion on shifting health priorities is abrupt. A connecting sentence would improve the flow.

Thank you for pointing out this issue. We have added a connecting sentence to bridge the transition between the demographic statistics and the discussion on shifting health priorities. This addition aims to enhance the flow and coherence of the text. The revised section now provides a smoother transition and improved continuity. The changes can be found in the updated manuscript In line 81-83, was added with the following paragraph:

“This demographic transition poses significant challenges for healthcare systems, which must adapt to meet the needs of older adults . They need to find effective strategies to extend health care and to respond to the needs of older adults ”

Insufficient Explanation: The claim that "mental health is even more crucial than physical health for older adults" is a significant assertion that needs more context or examples to substantiate why mental health is deemed more critical. Thank you for highlighting this important point. We have revised the sentence to provide additional context and examples to better substantiate the assertion that mental health is crucial for older adults. The revised sentence now includes specific evidence and references to support this claim. In line 88-89, was added with the following paragraph:

“According to Levinson and Kaplan, self-rated mental health is more important than self-rated physical health in predicting overall self-rated health ”

Citation Placement: The placement of citations could be more precise. For example, placing the citation immediately after the relevant statistics (e.g., after "In 2020, 13.5% of the global population was 60 years or older (2). Thank you for your insightful comment, reference number 2 belongs to the whole sentence, that is: According to statistics reported in 2020, 13.5 percent of the world's population was 60 years old or older and it is estimated that by 2050, one in every five people will be older adults

Clarity in Sampling Formula: The sample size formula is presented as a mathematical equation but isn't clearly formatted for readability. Additionally, there is no explanation of the terms used in the formula. Thank you for your valuable feedback. We have revised the description of the sample size calculation to provide a clearer and more detailed explanation. In line 134-141, was modified.

Inclusion and Exclusion Criteria: The inclusion criteria mention "diagnosed with hypertension," but it's unclear why this specific condition was chosen or how it relates to the study's focus on happiness, self-efficacy, and self-esteem. Thank you for your insightful comment. We appreciate the opportunity to clarify this point. The study was designed to include elderly participants regardless of their hypertension status, as we aimed to captu

---

## [Decision Letter · Decision Letter 1]

8 Oct 2024

PONE-D-24-19287R1Linking Self-Efficacy, Self-Esteem  and Happiness in Older People: A Cross-Sectional StudyPLOS ONE

Dear Dr. Maleki Chollou,

Thank you for submitting your manuscript to PLOS ONE. After careful consideration, we feel that it has merit but does not fully meet PLOS ONE’s publication criteria as it currently stands. Therefore, we invite you to submit a revised version of the manuscript that addresses the points raised during the review process.

We look forward to receiving your revised manuscript.

Kind regards,

Ayman Mohamed El-Ashry, Associate professor, Ph.D

Academic Editor

PLOS ONE

Additional Editor Comments:

The report titled "The Effect of Paradoxical Leadership on Nurse’s Performance: The Mediating Role of Thriving at Work" investigates how paradoxical leadership influences nurse performance, with thriving at work acting as a mediator. The study, conducted with 323 nurses, highlights the significance of leadership styles that balance competing demands, such as autonomy and control, in improving nurse performance.

Key findings indicate that paradoxical leadership positively correlates with nurse performance and thriving at work. Nurses who thrive, experiencing both learning and vitality, show better job performance, confirming the mediating role of thriving in the leadership-performance relationship. This suggests that fostering an environment where nurses can thrive is essential for maximizing performance, especially in demanding healthcare settings.

The study's implications suggest healthcare organizations should adopt leadership training focused on paradoxical leadership styles, promoting autonomy, learning, and vitality. Such programs could significantly enhance nurse performance and improve patient care quality. Future research is recommended to explore other mediating variables and use longitudinal designs to establish causality.

Reviewers' comments:

Reviewer's Responses to Questions

**Comments to the Author**

1. If the authors have adequately addressed your comments raised in a previous round of review and you feel that this manuscript is now acceptable for publication, you may indicate that here to bypass the “Comments to the Author” section, enter your conflict of interest statement in the “Confidential to Editor” section, and submit your "Accept" recommendation.

Reviewer #1: All comments have been addressed

Reviewer #2: All comments have been addressed

Reviewer #3: All comments have been addressed

2. Is the manuscript technically sound, and do the data support the conclusions?

Reviewer #1: Yes

Reviewer #2: Yes

Reviewer #3: Yes

3. Has the statistical analysis been performed appropriately and rigorously? 

Reviewer #1: Yes

Reviewer #2: Yes

Reviewer #3: Yes

4. Have the authors made all data underlying the findings in their manuscript fully available?

Reviewer #1: Yes

Reviewer #2: Yes

Reviewer #3: Yes

5. Is the manuscript presented in an intelligible fashion and written in standard English?

Reviewer #1: Yes

Reviewer #2: Yes

Reviewer #3: No

6. Review Comments to the Author

Reviewer #1: I have carefully reviewed the revised manuscript and can confirm that all the comments and suggestions provided during the initial review have been addressed appropriately. The authors have incorporated the necessary changes, and the manuscript has improved significantly as a result. I appreciate the efforts made to enhance the clarity and quality of the paper. Based on my evaluation, the revisions meet the expectations, and I believe the manuscript is now suitable for publication. Thank you for the opportunity to review this work

Reviewer #2: (No Response)

Reviewer #3: Title The current title is clear, but it could be more concise while still capturing the essence of the study. For example:

"Self-Efficacy, Self-Esteem, and Happiness in Older Adults: A Cross-Sectional Study"

Abstract Background: The abstract is generally clear, but adding a more explicit statement on the research gap would strengthen it.

Results: Include specific quantitative data to reinforce your claims. For example, when mentioning the hierarchical multiple regression analysis, include the exact values for the coefficients or percentages of variance explained.

Conclusion: Your conclusion mentions interventions, but this could be more specific. Instead of broadly suggesting “support groups,” you could suggest types of interventions that have been proven to work, like "skill-building workshops aimed at enhancing self-efficacy."

Introduction Specificity: Be more specific about the advancements in medical care and public health mentioned. Providing examples or citations can give more context. For instance, highlight advancements in particular areas like preventive medicine or technology that extend health span.

Research Gap: it would be beneficial to expand on why Iranian elderly rank low in happiness compared to global counterparts, perhaps by discussing specific cultural or socioeconomic factors unique to Iran.

Clarity: Some parts, especially around the discussion of cognitive factors feel a bit abstract. You can ground these concepts by referring to specific studies or providing more concrete examples.

Methods Sampling Method: The term "multi-stage cluster sampling" is mentioned, but the process could be better clarified. You might want to add a sentence or two explaining why this method was chosen and how it benefits the study. You could elaborate on how clusters were selected (e.g., health centers or regions), and how this sampling ensures representativeness.

Inclusion Criteria: It's great that you have clear inclusion/exclusion criteria, but it’s important to explain why participants with psychiatric disorders were excluded. What impact would their inclusion have on the study’s outcomes?

Data Collection: Adding a brief explanation about the measures taken to ensure that interviewers’ bias was minimized (e.g., through standardized scripts or blind interviews) could strengthen the validity of your methods.

Results Interpretation of Statistical Data: In Table 3, the values could be interpreted more clearly. You could explain how the changes in R² (e.g., from 0.153 to 0.587) reflect the significant role of self-efficacy and self-esteem beyond demographic factors.

Detailing Key Findings: For the significant predictors, such as self-efficacy and self-esteem, a brief explanation on why they contribute so heavily to happiness in your population could provide a deeper understanding of your results.

Discussion Causal Inferences: The discussion seems to make causal interpretations from a cross-sectional study. Since your study is correlational, emphasize that while relationships were found, causality cannot be inferred. Adding a statement to this effect in the discussion and limitations will strengthen the scientific rigor of your work.

Comparative Analysis: Compare your findings more explicitly with studies in different cultural or geographical settings. You cite many studies from other countries (e.g., Turkey, Thailand), but a more nuanced comparison of how these findings align with or differ from your results will add depth.

Mechanisms of Influence: In lines 266-268, you mention that increasing self-efficacy and self-esteem can enhance happiness. Discuss potential mechanisms for this influence. For instance, is it because of improved coping strategies, greater social participation, or reduced psychological distress?

Interventions: In lines 305-307, be more specific about the types of community-based programs you recommend. For example, "Programs incorporating cognitive-behavioral therapy or mindfulness training could effectively boost self-efficacy and self-esteem."

Limitations Expanded Limitations: You mention that the study’s cross-sectional design limits your ability to draw causal conclusions. It would be beneficial to mention other limitations, such as reliance on self-reported measures, which can introduce bias. Also, acknowledge any potential biases in sampling (e.g., if the study underrepresents a certain subgroup of the elderly population in Sarab).

Generalizability: The generalizability of your results outside the Iranian context could be addressed. You may want to elaborate on how cultural differences might affect self-efficacy, self-esteem, and happiness relationships in other contexts.

Conclusion Action-Oriented Suggestions: Your conclusion should not only summarize the findings but also provide clearer actionable recommendations for healthcare providers and policymakers. Rather than just suggesting “preventive and supportive actions,” you could specify programs that could be implemented, such as "targeted cognitive-behavioral programs aimed at boosting self-efficacy and emotional regulation in older adults."

Language and Formatting Clarity and Consistency: The manuscript can be improved by revising some repetitive phrases and ensuring consistency in terms (e.g., consistently using either "older adults" or "elderly" rather than switching between them). Also, ensure that all abbreviations are defined the first time they are used.

Grammar and Flow: In some sections, the flow of ideas is somewhat disrupted by repetitive transitions (e.g., lines 235-236). Try to vary sentence structure and reduce redundancy to improve readability.

Novelty Emphasize the unique contributions of your research. For example, highlight how your study provides new insights into an under-researched population (Iranian elderly) in the context of self-efficacy, self-esteem, and happiness. Stress this in both the introduction and conclusion to reinforce the study’s value.

7. PLOS authors have the option to publish the peer review history of their article (what does this mean? ). If published, this will include your full peer review and any attached files.

**Do you want your identity to be public for this peer review?** For information about this choice, including consent withdrawal, please see our Privacy Policy .

Reviewer #1: **Yes: ** shaimaa mohamed amin

Reviewer #2: No

Reviewer #3: **Yes: ** Eslam Reda Machaly

---

## [Author Response · Author response to Decision Letter 1]

11 Dec 2024

POINT-BY-POINT RESPONSE FORM

Title: Response to Reviewer Comments on Manuscript

To the Editor of Plos One : Dr Ayman Mohamed El-Ashry

Manuscript : PONE-D-24-19287

Dear Editor,

On behalf of my co-authors, I, Khalil Maleki Chollou, would like to express our sincere gratitude to you and the reviewers for the thorough review and valuable feedback on our manuscript . Your comments will greatly help us improve the quality of our paper.

In this letter, we intend to address each of the reviewers' comments and suggestions in detail.

Suggestion, Question,

or Comment from the

Reviewer #3 Author’s Response Change in the Manuscript

The current title is clear, but it could be more concise while still capturing the essence of the study. For example:

"Self-Efficacy, Self-Esteem, and Happiness in Older Adults: A Cross-Sectional Study" We truly appreciate the Reviewer #3’s suggestion, and we recognize that it would improve the revised version of the manuscript.

Change of Manuscript Title from “Linking Self-Efficacy, Self-Esteem and Happiness in Older People: A Cross-Sectional Study” to “Self-Efficacy, Self-Esteem, and Happiness in Older Adults: A Cross-Sectional Study”

Abstract Background: The abstract is generally clear, but adding a more explicit statement on the research gap would strengthen it. Thank you for this valuable suggestion. We have revised the Background section of the abstract to explicitly state the research gap by highlighting the lack of studies examining the combined effect of self-efficacy and self-esteem on happiness in elderly populations. This can be found in lines 29–34 of the revised abstract.

Abstract Results: Include specific quantitative data to reinforce your claims. For example, when mentioning the hierarchical multiple regression analysis, include the exact values for the coefficients or percentages of variance explained. Thank you for this valuable suggestion. We have revised the Results section of the abstract. This can be found in lines 42–48 of the revised abstract.

Abstract Conclusion: Your conclusion mentions interventions, but this could be more specific. Instead of broadly suggesting “support groups,” you could suggest types of interventions that have been proven to work, like "skill-building workshops aimed at enhancing self-efficacy." We appreciate this suggestion to provide more specific recommendations. We have revised the Conclusion to include more targeted interventions, such as "skill-building workshops" and "art therapy programs."

These changes are now reflected in lines 49–56 of the revised abstract.

Introduction Specificity: Be more specific about the advancements in medical care and public health mentioned. Providing examples or citations can give more context. For instance, highlight advancements in particular areas like preventive medicine or technology that extend health span. we have added specific examples of advancements in medical care, including improvements in preventive medicine and technological innovations like telemedicine and wearable health monitors, to provide clearer context. These changes are now reflected in lines 60–65 of the revised Introduction.

Introduction Research Gap: it would be beneficial to expand on why Iranian elderly rank low in happiness compared to global counterparts, perhaps by discussing specific cultural or socioeconomic factors unique to Iran. We have expanded the discussion on why Iranian seniors rank low in happiness by including cultural and socioeconomic factors unique to Iran, such as economic instability and limited social support networks.

This can be found in lines 102–106 of the revised Introduction

Introduction Clarity: Some parts, especially around the discussion of cognitive factors feel a bit abstract. You can ground these concepts by referring to specific studies or providing more concrete examples. We truly appreciate the Reviewer #3’s suggestion, and we recognize that it would improve the revised version of the manuscript.

Methods Sampling Method: The term "multi-stage cluster sampling" is mentioned, but the process could be better clarified. You might want to add a sentence or two explaining why this method was chosen and how it benefits the study. You could elaborate on how clusters were selected (e.g., health centers or regions), and how this sampling ensures representativeness.

Thank you for your insightful comments regarding the sampling method used in our study. In response to your suggestion for clarification, I have revised the description of the sampling method.

These changes are now reflected in lines 133–141 of the revised Methods.

Methods Inclusion Criteria: It's great that you have clear inclusion/exclusion criteria, but it’s important to explain why participants with psychiatric disorders were excluded. What impact would their inclusion have on the study’s outcomes? Thank you for your insightful comment regarding the inclusion criteria for participants in our study. We appreciate your attention to detail and the importance of clearly explaining our rationale for excluding individuals with psychiatric disorders.

In this study, participants were individuals aged 60 years or older, free from psychiatric disorders, and willing to sign a consent form to participate. The exclusion of individuals with psychiatric disorders was implemented for several reasons:

Minimizing Confounding Variables: Individuals with psychiatric disorders may experience different levels of happiness, self-efficacy, and self-esteem compared to those without such disorders. Their inclusion could introduce confounding variables that might skew the results and complicate the interpretation of the relationships between self-efficacy, self-esteem, and happiness.

Focus on Psychological Constructs: The primary focus of our study is to investigate the relationships between self-efficacy, self-esteem, and happiness in a generally healthy elderly population. By excluding individuals with psychiatric disorders, we aimed to ensure that our findings accurately reflect the influence of these psychological constructs in a population where other mental health factors do not significantly interfere.

Enhancing Validity: By creating a more homogeneous sample, we enhance the validity and reliability of our findings. This allows us to draw more precise conclusions about the associations we are investigating and to better inform interventions aimed at improving the well-being of older adults.

We hope that this explanation addresses your concerns and clarifies the rationale behind our inclusion and exclusion criteria.

Methods Data Collection: Adding a brief explanation about the measures taken to ensure that interviewers’ bias was minimized (e.g., through standardized scripts or blind interviews) could strengthen the validity of your methods. Thank you for this insightful comment. To address your concern, I have added further explanations regarding the measures implemented to minimize interviewer bias in the data collection section. These changes are now reflected in lines 150–157 of the revised Methods.

Results Interpretation of Statistical Data: In Table 3, the values could be interpreted more clearly. You could explain how the changes in R² (e.g., from 0.153 to 0.587) reflect the significant role of self-efficacy and self-esteem beyond demographic factors. We appreciate the reviewer’s insightful suggestion regarding the interpretation of Table 3. We have revised the Results section to provide a clearer explanation of the changes in R². Specifically, we now emphasize that the initial model with demographic factors explained 15.3% of the variation in happiness, indicating their foundational role. Adding self-efficacy and self-esteem in step 2 increased the explained variance to 58.7%, highlighting the critical contribution of these psychological factors. Self-efficacy (β = 0.695) was identified as the most significant predictor, underscoring its importance in understanding happiness among elders. These clarifications have been incorporated into the manuscript for greater clarity. These changes are now reflected in lines 227–236 of the revised Results.

Results Detailing Key Findings: For the significant predictors, such as self-efficacy and self-esteem, a brief explanation on why they contribute so heavily to happiness in your population could provide a deeper understanding of your results. We appreciate your insightful comment regarding the need to elaborate on the significant predictors of happiness, such as self-efficacy and self-esteem. In response, we have added an explanation in the Results section to clarify their roles. These changes are now reflected in lines 242–247 of the revised Results.

Discussion Causal Inferences: The discussion seems to make causal interpretations from a cross-sectional study. Since your study is correlational, emphasize that while relationships were found, causality cannot be inferred. Adding a statement to this effect in the discussion and limitations will strengthen the scientific rigor of your work. All the comments were made point-by-point and marked by track changes and highlighting in the revision file of the manuscript.

Discussion Comparative Analysis: Compare your findings more explicitly with studies in different cultural or geographical settings. You cite many studies from other countries (e.g., Turkey, Thailand), but a more nuanced comparison of how these findings align with or differ from your results will add depth. All the comments were made point-by-point and marked by track changes and highlighting in the revision file of the manuscript. The results of this research compare with studies in Tehran, Iran, Hamadan, Iran.

Discussion Mechanisms of Influence: In lines 266-268, you mention that increasing self-efficacy and self-esteem can enhance happiness. Discuss potential mechanisms for this influence. For instance, is it because of improved coping strategies, greater social participation, or reduced psychological distress? All the comments were made point-by-point and marked by track changes and highlighting in the revision file of the manuscript. It was added.

Discussion Interventions: In lines 305-307, be more specific about the types of community-based programs you recommend. For example, "Programs incorporating cognitive-behavioral therapy or mindfulness training could effectively boost self-efficacy and self-esteem." All the comments were made point-by-point and marked by track changes and highlighting in the revision file of the manuscript. It was added.

Limitations Expanded Limitations: You mention that the study’s cross-sectional design limits your ability to draw causal conclusions. It would be beneficial to mention other limitations, such as reliance on self-reported measures, which can introduce bias. Also, acknowledge any potential biases in sampling (e.g., if the study underrepresents a certain subgroup of the elderly population in Sarab). These limitations have been added.

Limitations Generalizability: The generalizability of your results outside the Iranian context could be addressed. You may want to elaborate on how cultural differences might affect self-efficacy, self-esteem, and happiness relationships in other contexts. All the comments were made point-by-point and marked by track changes and highlighting in the revision file of the manuscript. It was corrected.

Conclusion Action-Oriented Suggestions: Your conclusion should not only summarize the findings but also provide clearer actionable recommendations for healthcare providers and policymakers. Rather than just suggesting “preventive and supportive actions,” you could specify programs that could be implemented, such as "targeted cognitive-behavioral programs aimed at boosting self-efficacy and emotional regulation in older adults." Thank you for this insightful suggestion. We have revised the conclusion to not only summarize our findings but also to include more specific, actionable recommendations for healthcare providers and policymakers.

This can be found in lines 342–360 of the revised Conclusion.

Language and Formatting

Clarity and Consistency: The manuscript can be improved by revising some repetitive phrases and ensuring consistency in terms (e.g., consistently using either "older adults" or "elderly" rather than switching between them). Also, ensure that all abbreviations are defined the first time they are used.

Grammar and Flow: In some sections, the flow of ideas is somewhat disrupted by repetitive transitions (e.g., lines 235-236). Try to vary sentence structure and reduce redundancy to improve readability. All the comments were made point-by-point and marked by track changes and highlighting in the revision file of the manuscript. It was added.

Novelty:Emphasize the unique contributions of your research. For example, highlight how your study provides new insights into an under-researched population (Iranian elderly) in the context of self-efficacy, self-esteem, and happiness. Stress this in both the introduction and conclusion to reinforce the study’s value. In conclusion and discussion sections, new insights have be added

With the changes outlined in this letter, we are confident that the manuscript is scientifically richer and of higher quality.

We thank you and the reviewers for your patience and thorough review of this manuscript.

Sincerely,

Khalil Maleki Chollou

---

## [Decision Letter · Decision Letter 2]

30 Jan 2025

Self-Efficacy, Self-Esteem, and Happiness in Older Adults: A Cross-Sectional Study

PONE-D-24-19287R2

Dear Dr. Maleki Chollou,

We’re pleased to inform you that your manuscript has been judged scientifically suitable for publication and will be formally accepted for publication once it meets all outstanding technical requirements.

Kind regards,

Associate Professor Dr. Nik Ahmad Sufian Burhan

Academic Editor

PLOS ONE

Additional Editor Comments (optional):

Reviewers' comments:

Reviewer's Responses to Questions

**Comments to the Author**

1. If the authors have adequately addressed your comments raised in a previous round of review and you feel that this manuscript is now acceptable for publication, you may indicate that here to bypass the “Comments to the Author” section, enter your conflict of interest statement in the “Confidential to Editor” section, and submit your "Accept" recommendation.

Reviewer #3: All comments have been addressed

2. Is the manuscript technically sound, and do the data support the conclusions?

Reviewer #3: Yes

3. Has the statistical analysis been performed appropriately and rigorously? 

Reviewer #3: Yes

4. Have the authors made all data underlying the findings in their manuscript fully available?

Reviewer #3: Yes

5. Is the manuscript presented in an intelligible fashion and written in standard English?

Reviewer #3: Yes

6. Review Comments to the Author

Reviewer #3: (No Response)

7. PLOS authors have the option to publish the peer review history of their article (what does this mean? ). If published, this will include your full peer review and any attached files.

**Do you want your identity to be public for this peer review?** For information about this choice, including consent withdrawal, please see our Privacy Policy .

Reviewer #3: **Yes: ** Eslam Reda Machaly

---

## [Editor Report · Acceptance letter]

PONE-D-24-19287R2

PLOS ONE

Dear Dr. Maleki Chollou,

I'm pleased to inform you that your manuscript has been deemed suitable for publication in PLOS ONE. Congratulations! Your manuscript is now being handed over to our production team.

Kind regards,

on behalf of

Dr. Nik Ahmad Sufian Burhan

Academic Editor

PLOS ONE